# RNA helicase EIF4A1-mediated translation is essential for the GC response

Michael Screen[1], Louise S Matheson[1], Andrew JM Howden[2], Douglas Strathdee[3], Anne E Willis[4], Martin Bushell[3,5], Owen Sansom[3,5], Martin Turner[1]

**EIF4A1 and cofactors EIF4B and EIF4H have been well characterised in cancers, including B cell malignancies, for their ability to promote the translation of oncogenes with structured 5′ untranslated regions. However, very little is known of their roles in nonmalignant cells. Using mouse models to delete *Eif4a1*, *Eif4b* or *Eif4h* in B cells, we show that EIF4A1, but not EIF4B or EIF4H, is essential for B cell development and the germinal centre response. After B cell activation in vitro, EIF4A1 facilitates an increased rate of protein synthesis, MYC expression, and expression of cell cycle regulators. However, EIF4A1-deficient cells remain viable, whereas inhibition of EIF4A1 and EIF4A2 by Hippuristanol treatment induces cell death.**

## Introduction

Eukaryotic initiation factor (EIF)-4F is a multiprotein complex, which facilitates the loading of mRNA onto the small ribosomal subunit pre-initiation complex (PIC). It is comprised of the EIF4G scaffold protein; EIF4E, which binds to the mRNA cap; and EIF4A, an ATP-driven RNA helicase. The ATPase activity of EIF4A is required for mRNA loading onto the PIC and unwinds the 5′ untranslated region (UTR) of mRNA to facilitate scanning by the PIC. The helicase activity of EIF4A by itself is weak but is strongly stimulated in the presence of cofactors EIF4B or EIF4H, EIF4G, and purine-rich RNA (1, 2). The mTOR pathway can activate EIF4A through promoting EIF4B function (3) or degradation of PDCD4, a negative regulator of EIF4A (4). EIF4A can promote the translation of mRNAs with highly structured 5′UTRs, a feature of many oncogenes. Enhanced EIF4A activity is associated with malignant proliferation and the survival of tumour cells (5, 6, 7), and small-molecule inhibitors of EIF4A have been developed as anti-cancer drugs (7, 8, 9, 10, 11).

Two paralogs of EIF4A, EIF4A1 and EIF4A2, are 90% identical at the amino acid level, but appear to be functionally distinct with EIF4A1 associated with growth and proliferation and EIF4A2 associated

with quiescence. In cell-free systems, EIF4A2 can perform the same functions as EIF4A1, but rarely compensates for the loss of EIF4A1 in cancer cell lines (12). Little is known about these in primary tissue: both *Eif4a1* and *Eif4b* are essential for mouse development (13, 14); mice with conditional *Eif4b* deletion in adulthood had increased mortality and susceptibility to viral infection (13); *Eif4a2* and *Eif4h* knock-out mice are viable, but have growth and developmental abnormalities (14, 15). In mouse lymphoma models, the loss of a single copy of *Eif4a1* or *Eif4e* by the tumour leads to improved survival of the host (14, 16). Although EIF4A1 is known to be induced in B cells by B cell receptor signaling, the roles of EIF4A1 and its associated factors EIF4B and EIF4H in B cell development and activation have not been studied.

We address this using mouse models to conditionally inactivate *Eif4a1*, *Eif4b*, and *Eif4h* during B cell development and activation. Our results show EIF4A1 is required for B cell development and the germinal centre (GC) response but not for the maintenance of mature B cells. EIF4B and EIF4H appear to be dispensable for B cell development and activation except when both are deleted in a competitive environment. EIF4A1-deficient naïve B cells have a normal rate of translation, but naïve B cells treated with hippuristanol, which inhibits both EIF4A1 and EIF4A2, show reduced protein synthesis and increased cell death. Furthermore, we demonstrate that EIF4A1 becomes essential for the increase in translation after B cell activation. These results indicate an indispensable requirement for EIF4A1 in B cell development and activation.

## Results

### Developing B cells require EIF4A1 but not cofactors EIF4B or EIF4H

To investigate the role of EIF4A1 in B cell development, we deleted *Eif4a1* in pro-B cells using *Cd79a-cre*. *Eif4a1^{fl/fl} Cd79a^{cre/+}* mice had an almost complete loss of splenic B cells (Fig 1A). In the bone marrow, *Eif4a1^{fl/fl} Cd79a^{cre/+}* mice had normal numbers of

[1]Immunology Programme, The Babraham Institute, Babraham Research Campus, Cambridge, UK    [2]Cell Signalling and Immunology, University of Dundee, Dundee, UK    [3]Cancer Research UK Beatson Institute, Glasgow, UK    [4]MRC Toxicology Unit, University of Cambridge, Cambridge, UK    [5]School of Cancer Sciences, University of Glasgow, Glasgow, UK

Correspondence: michael.screen@babraham.ac.uk; martin.turner@babraham.ac.uk

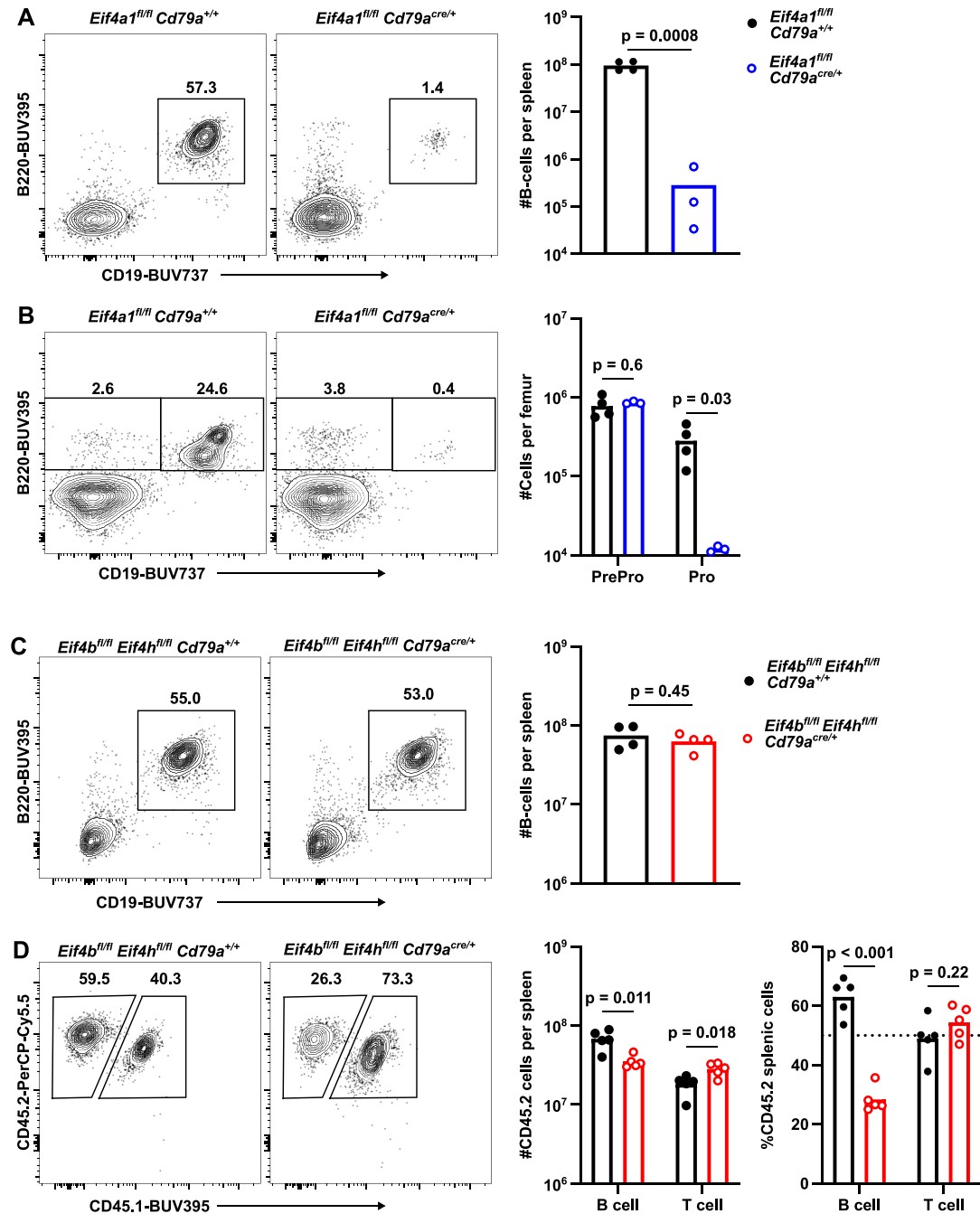

**Figure 1. B cell development requires EIF4A1 but not EIF4B or EIF4H.**
**(A, B, C, D)** Left: representative flow cytometry plots showing gating strategy on events pre-gated for viable cells (A, B, C) or CD19⁺ cells (D); numbers indicate the percentage of the gated population. **(A, B, C, D)** Right: (A) number of splenic CD19⁺ B220⁺ cells; (B) number of Pre-Pro (CD19⁻ B220⁺) and Pro (CD19⁺ B220+ Igμ⁻) B cells; (C) number of splenic CD19⁺ B220⁺ cells; (D) number and percentage of CD45.2⁺ CD45.1⁻ in B (CD19⁺) and T (TCRB⁺) cells. In (B, C) the data are representative of two independent experiments. All graphs show data from individual mice with bar charts representing the mean.

B220+ CD19⁻ (pre-pro) or Fraction A cells using the Hardy classification (17) but a greater than 20-fold decrease in pro B cells (CD19⁺ Igμ−) or Hardy Fraction B (Figs 1B and S1A). The remaining pro B cells had only a minor reduction in EIF4A1 expression and a 10-fold increase in EIF4A2 expression (Fig S1B) indicating that EIF4A2 cannot compensate for the loss of EIF4A1 in pro B cells. We interpret this to indicate that the remaining pro B cells had undergone CRE-

mediated recombination at the *Eif4a1* locus but have not yet degraded all of the EIF4A1 protein.

The helicase activity of EIF4A1 can be significantly stimulated by the interaction with EIF4B or EIF4H. To understand whether these cofactors are required in developing B cells, we deleted both *Eif4b* and *Eif4h* during B cell development. *Eif4b*ᶠˡ/ᶠˡ *Eif4h*ᶠˡ/ᶠˡ *Cd79a*ᶜʳᵉ/⁺ mice showed efficient loss of both proteins (Fig S1C) and had

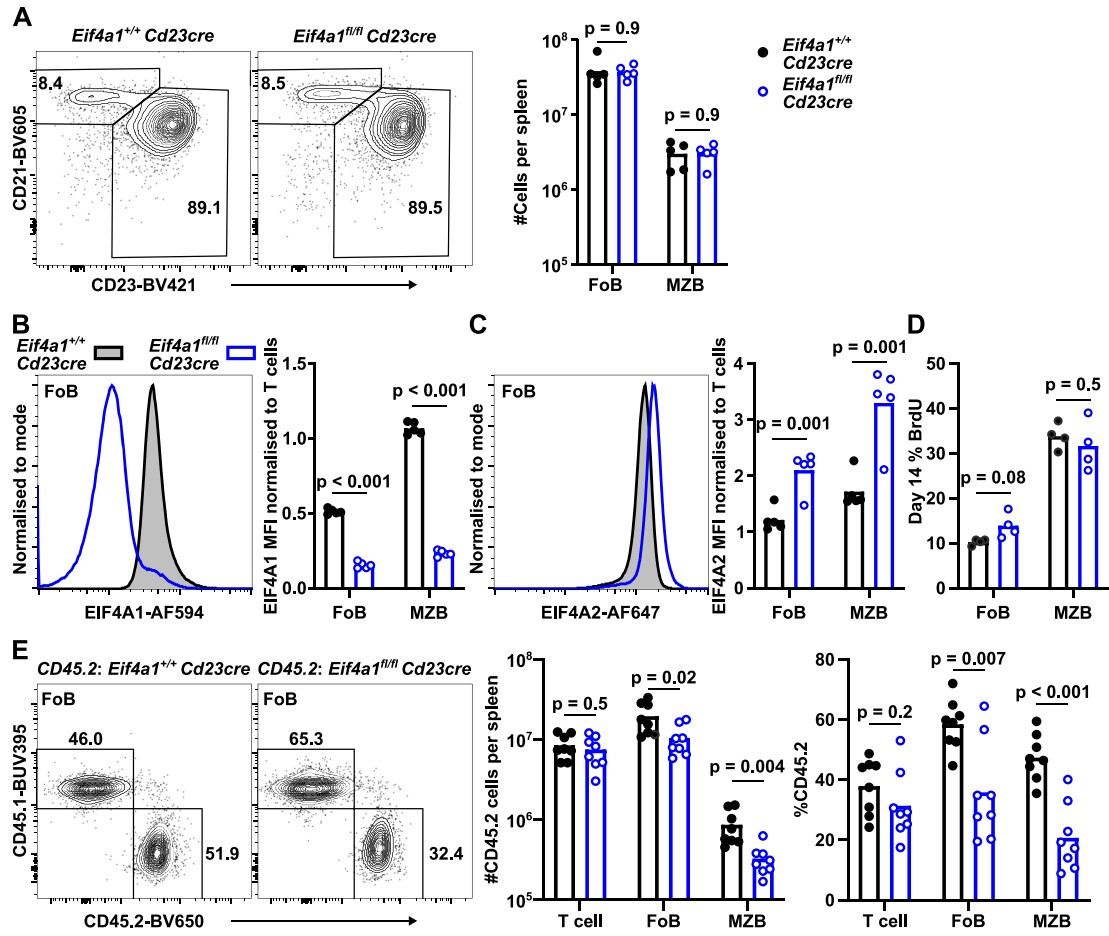

**Figure 2. EIF4A1 is not essential for mature B cells.**
**(A, E)** Left: representative flow cytometry plots showing gating strategy on events pre-gated for CD19⁺ CD93⁻ (A) or CD19⁺ CD23⁺ CD21⁺ (E); numbers indicate the percentage of the gated population. **(A, E)** Right: (A) number of follicular (Fo, CD23⁺ CD21⁺) and marginal zone (MZ, CD23ˡᵒ CD21⁺); (E) number and percentage of CD45.2⁺ CD45.1⁻ T-cells (TCRB⁺), Fo and MZ B cells. **(B, C)** Flow cytometry analysis of EIF4A1 (B) and EIF4A2 (C) expression from cells in (A). Left: representative flow cytometry plots. Right: graphs show median fluorescence intensity and are representative of two independent experiments. **(D)** Percentage of BrdU in Fo and MZ B cells from mice treated with BrdU for 14 d. All graphs show data from individual mice with bar charts representing the mean.

normal numbers of splenic B cells (Fig 1C). Enumeration of B cell developmental stages in the bone marrow showed normal numbers of fractions B–E and only a small (25%) reduction in mature/recirculating B cells (Fig S1D). In a mixed bone marrow chimera, the number of *Eif4b^{fl/fl} Eif4h^{fl/fl} Cd79a^{cre/+}* splenic B cells is decreased by 48% compared with mice that received *Eif4b^{fl/fl} Eif4h^{fl/fl} Cd79a^{+/+}* bone marrow cells (Fig 1D). Therefore, EIF4A1 is essential at the pro B cell stage, but its cofactors EIF4B or EIF4H are not required at any stage in B cell development. We conclude that the combined function of EIF4B and EIF4H promotes the competitive fitness of B cells during development.

## Maintenance of mature B cells is not EIF4A1-dependent

To bypass the developmental block in *Eif4a1^{fl/fl} Cd79a^{cre/+}* mice, we crossed *Eif4a1^{fl/fl}* mice to *Cd23-cre* mice, which results in deletion of *Eif4a1* in splenic immature B cells. *Cd23-cre*–mediated deletion of *Eif4a1* had no impact on follicular (Fo) or marginal zone (MZ) B cell numbers in the spleen (Fig 2A). Fo and MZ B cells are estimated to express of 3.3 × 10⁵ and 7 × 10⁵ copies of EIF4A1 and 9 × 10⁴ and 1.3 × 10⁵ copies of EIF4A2, respectively (18). In *Eif4a1^{fl/fl} Cd23-cre* mice

both Fo and MZ B cells had a substantial loss of EIF4A1 expression (Fig 2B) and had a twofold increase in EIF4A2 expression (Fig 2C). We assessed B cell turnover by labelling newly produced B cells in the bone marrow with bromodeoxyuridine (BrdU) for 14 d. Both Fo and MZ B cells from *Eif4a1^{fl/fl} Cd23-cre* mice had similar BrdU incorporation compared with *Eif4a1^{+/+} Cd23-cre* mice (Fig 2D). Loss of EIF4A1 therefore does not obviously affect the lifespan of mature B cells. In a mixed bone marrow competitive chimera, the number of Fo and MZ cells derived from *Eif4a1^{fl/fl} Cd23-cre* bone marrow was reduced by 47% and 59%, respectively, compared with mice that received *Eif4a1^{+/+} Cd23-cre* bone marrow cells (Fig 2E). This indicates a minor role for EIF4A1 in the formation or persistence of Fo and MZ cells. Overall, it is clear that EIF4A1 is not essential to maintain mature quiescent B cells.

## EIF4A1 is required for germinal centre formation and antibody response

The transition from a naïve to germinal centre (GC) B cell leads to an increased expression of EIF4A1 (3.4-fold), EIF4B (2.3-fold), and EIF4H (3.6-fold) and a 50% decrease in EIF4A2 expression (Fig S2A and B).

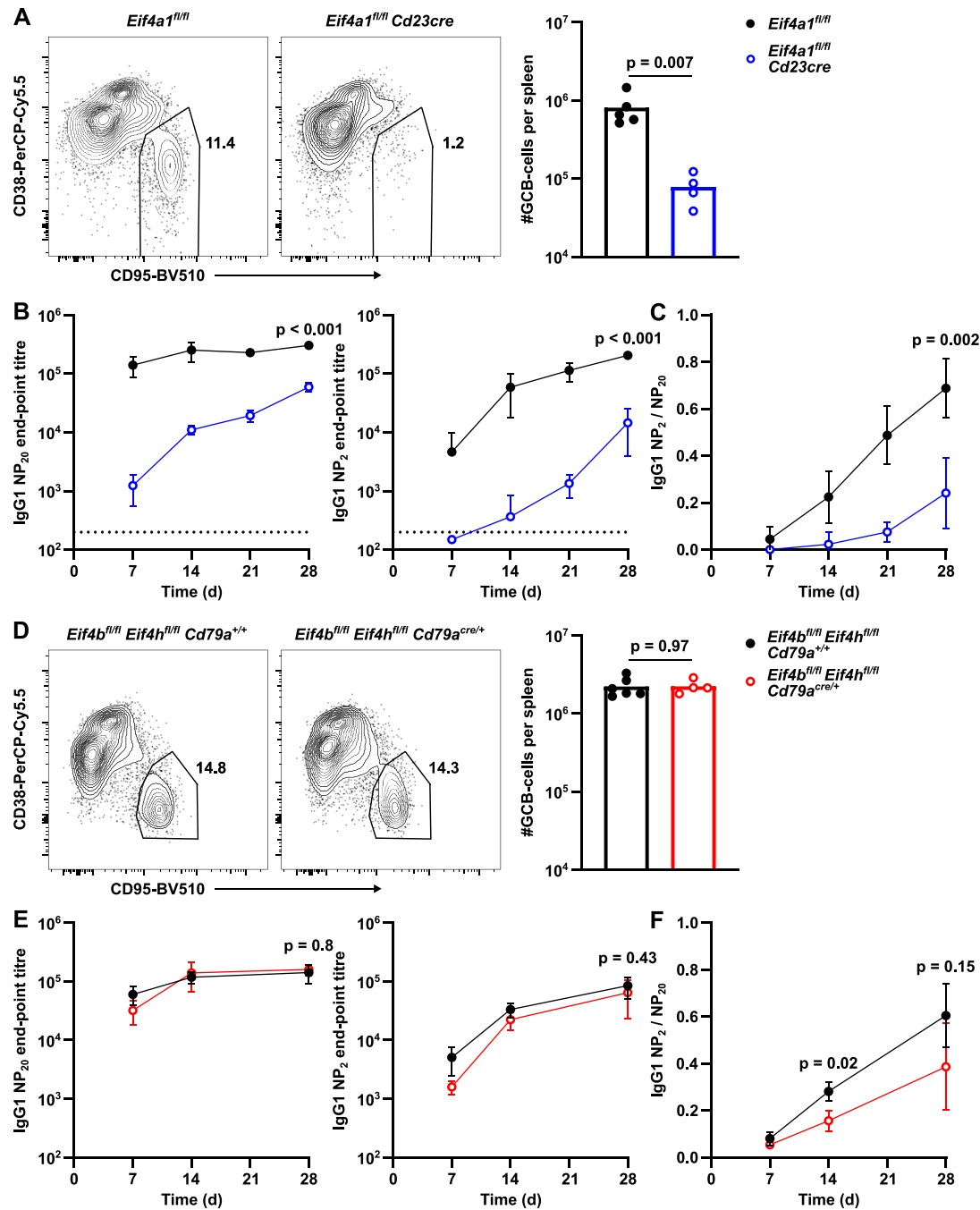

**Figure 3. EIF4A1, but not EIF4B or EIF4H, is required in B cells for a GC response.**
**(A, D)** Left: representative flow cytometry plots showing gating strategy on events pre-gated for CD19+ IgD-cells; number indicates the percentage of the gated population. Right: number of GCB cells. Data are representative of two independent experiments. **(B, E)** IgG1 endpoint titres measured by ELISA in sera of mice for up to 28 d after NP-KLH immunisation. Anti-NP20 (high and low affinity) or anti-NP2 (high affinity only). Data show the mean of 5 mice ± SD. **(C, F)** Ratio of IgG1 high affinity (NP2) to high and low affinity (NP20) measured from data shown in B (C) and D (E). Data show the mean of five mice ± SD. *P*-values calculated for each timepoint with *t* test and Holm–Šídák multiple testing, representative *P*-values shown.

To investigate the roles of EIF4A1 in the humoral immune response, we immunised *Eif4a1*$^{fl/fl}$ *Cd23-cre* and *Eif4a1*$^{fl/fl}$ mice with 4-hydroxy-3-nitrophenyl-acetyl conjugated to keyhole limpet hemocyanin (NP-KLH) precipitated in alum. 7 d later, GC B cells were almost completely absent in *Eif4a1*$^{fl/fl}$ *Cd23-cre* mice with a 10-fold reduction in the

number of GC B cells in the spleen compared with EIF4A1-sufficient mice (Fig 3A). The number of light zone (LZ) and dark zone (DZ) GC B cells were reduced by 9-fold and 15-fold, respectively. In the GC, there was only a small reduction in the ratio of DZ to LZ cells from 1.5 in *Eif4a1*$^{fl/fl}$ mice to 1.0 in *Eif4a1*$^{fl/fl}$ *Cd23-cre* mice (Fig S2C).

The reduction in GC B cells corresponded with a greater than 100-fold reduction in NP-specific IgG1 antibody-secreting cells in the spleen (Fig S2D). Moreover, 28 d after NP-KLH immunisation there were fewer low- and high-affinity bone marrow antibody secreting cells in *Eif4a1*^fl/fl *Cd23-cre* mice compared with *Eif4a1*^fl/fl mice (Fig S2E). Furthermore, IgG1 antibodies of both low- and high-affinity were reduced in *Eif4a1*^fl/fl *Cd23-cre* mice compared with *Eif4a1*^fl/fl mice (Fig 3B). In *Eif4a1*^fl/fl mice, the proportion of high-affinity antibodies increased over time, but this remained low in *Eif4a1*^fl/fl *Cd23-cre* mice (Fig 3C). The remaining GC B cells in *Eif4a1*^fl/fl *Cd23-cre* mice had only a 40% reduction in EIF4A1 expression compared with GC B cells in *Eif4a1*^fl/fl mice (Fig S2A). Furthermore, expression of EIF4A1 was 2.1-fold increased and EIF4A2 30% reduced compared with EIF4A1-sufficient naïve B-cells, which suggest that some of the apparent response to NP-KLH immunisation in *Eif4a1*^fl/fl *Cd23-cre* mice is from partial escapees.

*Eif4b*^fl/fl *Eif4h*^fl/fl *Cd79a*^cre/+ mice immunised with NP-KLH had normal numbers of splenic GC B cells on day 7 (Fig 3D), despite efficient loss of both EIF4B and EIF4H (Fig S2B). *Eif4b*^fl/fl *Eif4h*^fl/fl *Cd79a*^cre/+ mice also produced both low- and high-affinity IgG1 antibodies (Fig 3E). A small reduction in the contribution of high-affinity antibody to the response of *Eif4b*^fl/fl *Eif4h*^fl/fl *Cd79a*^cre/+ compared with *Eif4b*^fl/fl *Eif4h*^fl/fl *Cd79a*^+/+ mice was only significant at 14 d after immunisation (Fig 3F). These data show B cell activation and/or the GC response are dependent upon EIF4A1 but not EIF4B or EIF4H in B cells.

### EIF4A1 links B cell activation with increased rate of translation

Activation of B cells via the B cell receptor, CD40, or the toll-like receptor leads to extensive remodelling of the B cell proteome, which promotes growth and proliferation. To understand whether the inability of *Eif4a1*^fl/fl *Cd23-cre* mice to mount a GC response was because of an activation defect, we isolated B cells from CD45.1 B6.SJL mice (control) and CD45.2 *Eif4a1*^fl/fl *Cd23-cre* mice (4A1 KO) and activated them in co-culture on a CD40 ligand and hBAFF expressing cell line (CD40LB) in the presence of IL4. 24-h after coculture with CD40LB expression of CD69, a marker of activation, was increased equivalently in both control and 4A1 KO B cells (Fig 4A) indicating that EIF4A1 is not required for the cells to sense CD40L. The percentage of CD45.2 4A1 KO B cells remained unchanged after 24 h culture suggesting that EIF4A1 was not required for survival during early activation (Fig 4B). Furthermore, *Eif4a1*^fl/fl *Cd23-cre* and CD45.1 B6.SJL B cells had a similar percentage of membrane permeable/active caspase 3+ cells (Fig 4C). EIF4A1 is thus not necessary for activated cells to remain viable for at least the first 24 h of activation in vitro.

After CD40LB activation B cells increase BCL-XL expression (19). In CD40LB-activated 4A1 KO cells, BCL-XL expression, as measured by flow cytometry, is increased 2.6-fold compared with IL4 only and is only reduced by 18% compared with EIF4A1-sufficient B cells (Fig 4D). The transcription factor MYC, which is essential for the GC reaction, has been widely cited as being EIF4A1-sensitive (6, 7, 20). We therefore also measured MYC expression by intracellular flow cytometry. The median fluorescent intensity of MYC in activated 4A1 KO cells was increased threefold compared with B cells cultured in IL4-only and was decreased by 21% in 4A1 KO B cells compared with EIF4A1-

sufficient B cells after CD40LB activation (Fig 4E). To investigate whether loss of EIF4A1 altered the global rate of translation, we measured the amount of puromycin incorporation in the final 10 min of a 24-h culture. In IL4-only conditions, puromycin incorporation in control and 4A1 KO B cells was the same (Fig 4F). However, the fivefold increase in puromycin incorporation after activation in control B cells was not seen in the 4A1 KO cells (Fig 4F). Therefore, whereas activated 4A1 KO B cells can express new proteins, such as CD69, MYC and BCL-XL, their rate of protein synthesis is substantially limited.

Hippuristanol is an EIF4A-specific inhibitor of translation that targets both EIF4A1 and EIF4A2 (21). After hippuristanol treatment, B cells from B6.SJL mice cultured in IL4 only, or with CD40LB + IL4, showed a dose-dependent increase in cell death at 24 h (Fig 4G). Furthermore, at the higher doses of hippuristanol, puromycin incorporation was almost completely lost in the remaining viable cells (Fig 4H). Together, these results show that whereas EIF4A1 is indispensable for B cells to increase their translation rate after activation, EIF4A2 may be able to partially compensate and allow B cell survival.

### The transition from quiescent to proliferative state is EIF4A1-dependent

To characterise the impact of EIF4A1 depletion on the transcriptome and proteome of activated B cells, we performed proteomic and RNA-seq analyses on paired 24-h CD40LB-activated B cells from *Eif4a1*^fl/fl *Cd23-cre* (4A1 KO) and *Eif4a1*^+/+ *Cd23-cre* (control) mice. At this early timepoint, the transcriptome of 4A1 KO B cells was significantly different to control B cells with 18% of the transcripts detected differentially expressed (Fig 5A and Table S1). The differences in the proteomes showed a global reduction in protein expression (Fig 5B and Table S2) with around 50% of the proteins detected in control B cells having decreased expression in 4A1 KO B cells.

We performed gene set enrichment analysis (GSEA) on the RNA-seq data using a list of genes with increased expression 2 h after B cell stimulation with anti-IgM or LPS (early response) and a list of genes with increased expression after 24 h of anti-IgM stimulation (late response) (Table S3). In 4A1 KO B cells, the early response genes were enriched as increased in expression, whereas late response genes were decreased in expression (Fig 5C). As 4A1 KO B cells have not progressed beyond an immediate activation state, they are likely to have a proliferation defect. This was evident from GSEA of the proteomic data using Hallmark gene sets (22), which identified cell cycle regulation (G2M checkpoint, E2F targets, and mitotic spindle) among the most decreased pathways in 4A1 KO B cells (Fig 5D and Table S4).

4A1 KO activated B cells had similar amounts of Ccnd2 mRNA, encoding the G1 cyclin, compared with control cells, but a greater than 60% reduction of Ccne1, Ccna2, and Ccnb1 mRNAs which encode cyclins involved from the late G1 phase (Fig 5E). This suggested that 4A1 KO B cells might be unable to progress past the G1 checkpoint. This was confirmed by cell cycle analysis after EdU uptake, which identified the percentage of cells in S-phase 24 h after CD40LB activation was reduced from 14% in control B cells to 1% in 4A1 KO B cells (Fig 5F). Furthermore, flow cytometry analysis at 72 h showed CellTrace-violet–labelled naïve 4A1 KO B cells had undergone significantly fewer divisions with over 70% of cells

**Figure 4. EIF4A1 facilitates increased translation after activation.**
B cells isolated from B6.SJL WT CD45.1 (Control, closed circles) and *Eif4a1*^fl/fl^ *Cd23-cre* (4A1 KO, open blue circles) mice co-cultured in IL4 with or without CD40LB-expressing feeder cells for 24 h. **(A, D, E, F)** Left: representative flow cytometry plots. Right: median fluorescence intensity, MFI, of CD69 (A); BCL-XL (D); MYC (E); and puromycin incorporation (F) by flow cytometry. **(B, C)** Percentage of viable CD45.2 cells (B) or cell permeable or active caspase 3+ (C) B cells. **(G, H)** Percentage of viable B cells (G) and puromycin incorporation (H) in B cells from B6.SJL mice treated with different doses of hippuristanol at the start of the culture and puromycin in the final 10 min of culture. Data show the mean of three mice ± SD.

remaining undivided compared with less than 10% in control cells. This led to a reduction in the average number of divisions, calculated using the proliferation index, from 3 in control cells to 1 in EIF4A1-deficient cells (Fig 5G). Progression of cells through G1 is controlled by the activity of CDK4 and CDK6, whereas the mRNA of

these cyclin-dependent kinases is not different between 4A1 KO and control B cells both proteins were reduced by more than twofold (Fig 5H). Overall, these results show that EIF4A1 promotes the expression of proteins involved in the cell cycle and allows activated B cells to undergo multiple cell divisions.

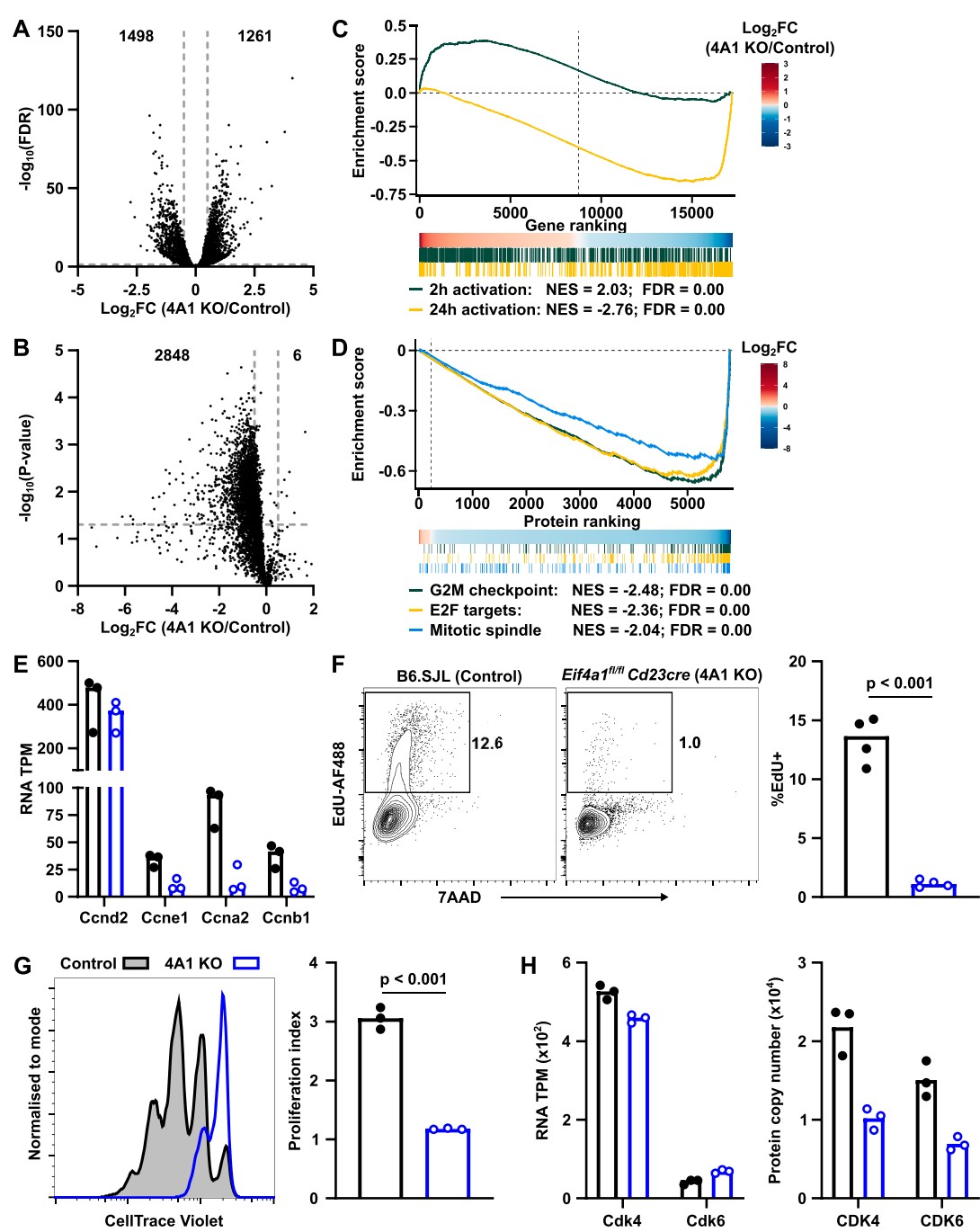

**Figure 5. EIF4A1 promotes B cell proliferation.**
**(A, B)** Log$_2$ fold change (x-axis) for (A) RNA-seq plotted against adjusted P-value (−log$_{10}$) and (B) Proteomics plotted against *P*-value (−log$_{10}$) of *Eif4a1$^{fl/fl}$ Cd23-cre* and *Eif4a1$^{+/+}$ Cd23-cre* B cells cultured with IL4 on CD40LB for 24 h. **(C)** GSEA for gene expression ranked by log$_2$FC using gene sets comprised of genes with increased expression 2 and 24 h after IgM stimulation. Normalised enrichment score and FDR, false discovery rate q-value shown. **(D)** GSEA for protein expression ranked by log$_2$FC using Hallmark gene sets. **(E)** Cyclin gene expression, transcript per million, TPM. **(F, G)** Flow cytometry analysis of cell cycle 24 h of co-culture on CD40LB cells based on incorporation of EdU with a fluorescent DNA-intercalator, 7-AAD (F). CellTrace-Violet dilution after 72 h of co-culture on CD40LB cells (G). B6.SJL (Control, closed circles) and *Eif4a1$^{fl/fl}$ Cd23-cre* (4A1 KO, open blue circles) B cells. Proliferation index calculated with FlowJo. **(H)** *Cdk4* and *Cdk6* expression. Left: RNA, TPM. Right: protein, Copy number. All bar graphs show data from individual mice with bar representing the mean.

# Discussion

Here, we present evidence for an essential role of EIF4A1 in B cells during B cell development and the germinal centre response.

EIF4A1 was not required to maintain quiescent mature B cells or for the early stages of B cell activation in vitro. We found that although EIF4A1-deficient B cells had a normal rate of protein synthesis, they were unable to increase their translation rate after activation.

Previous studies have shown the EIF4F factors, including EIF4A1, operate in a feedforward loop in which MYC increases transcription of *Eif4a1*, *Eif4e*, and *Eif4g1* and expression of these factors leads to increased MYC translation (23). In B and T cells, the amount of MYC after activation determines the capacity for cell division (24, 25). We found that whereas EIF4A1-deficient B cells had a reduced number of cell divisions after activation, this was unlikely to be driven solely because of the small decrease in MYC expression. EIF4A inhibition in cancer cells leads to G1-arrest through translation inhibition of CDK4, CDK6 or members of the cyclin D family (26, 27). Activated EIF4A1-deficient B cell protein but not RNA expression of CDK4 and CDK6 was reduced, which is likely to contribute to their arrest in G1.

However, given that half of the proteins detected by proteomics had reduced expression, it is likely EIF4A1 may also drive proliferation, in part, by promoting global protein synthesis.

In many cell types, the loss of EIF4A1 leads to increased EIF4A2 expression (12, 28, 29). In cancer cell lines, increased EIF4A2 expression does not compensate for the loss of EIF4A1 (12). EIF4A2 is increased in EIF4A1-deficient B cells and may have promoted survival of naïve B cells, but could not compensate for EIF4A1's role in proliferation. Hippuristanol treatment, which can inhibit both EIF4A1 and EIF4A2, increased cell death and reduced translation below that of EIF4A1-deficient cells. We propose that in primary B cells, and potentially other nonmalignant cell types (28), EIF4A2 may partially compensate for the loss of EIF4A1 by maintaining a basal level of protein synthesis sufficient for cell viability. *Eif4a2* however is not increased by MYC (23) and, even though it can participate in the EIF4F complex (12, 30), it can also associate with other proteins involved in suppression of translation (30, 31). This could explain why EIF4A2 cannot fully compensate for loss of EIF4A1, especially in highly proliferative cell types with high metabolic demands such as GC B cells, which share these features with cancer cells.

The activity of EIF4A is stimulated in the presence of cofactors including EIF4B and EIF4H. High expression of EIF4B has been shown to associate with poor prognosis in DLBCL (5) and its depletion leads to reduced growth of DLBCL cell lines (32). However, tamoxifen-mediated deletion of *Eif4b* in all cells of adult mice had no obvious effect on the proportion of B cells (13). It was therefore interesting that neither cofactor was essential for B cell development or the germinal centre response. A requirement for EIF4B and EIF4H was only evident when both were lost in a competitive environment, which is a stringent test of functionality. Thus, in the mouse, EIF4B and EIF4H appear to be dispensable in developing B cells and GC B cells but EIF4A1 has an essential role.

# Materials and Methods

## Mice

All mice were on a C57BL/6 background. *Eif4a1*-floxed mice, *Eif4a1*[tm2cBea], were generated by targeting mouse *Eif4a1* gene in HM1 ES cells. F1 offspring were crossed to actin-Flp and backcrossed to C57BL/6 for at least 10 generations. The *Eif4a1*[tm2cBea] allele places loxP sites on either side of exons 2 to 4, Ensembl transcript ID:

*Eif4a1*-213. *Eif4b*-floxed mice, *Eif4b*[tm1Tnr], were generated by targeting mouse *Eif4b* gene in C57BL/6J-Tyrc-2J/J ES cells. F1 offspring were crossed to C57BL/6 Flp deleter mice. The *Eif4b*[tm1Tnr], places loxP sites at either side of exons 3 and 4, Ensembl Transcript ID:*Eif4b*-201. *Eif4h*-floxed mice were generated by crossing *Eif4h*[tm1a(EUCOMM)Wtsi] mice with a flp recombinase expression mouse and have been and backcrossed to C57BL/6 for at least 10 generations. The allele places loxP sites on either side of exons 2 to 5, Ensembl Transcript ID: *Eif4h*-205. The abovementioned mice were crossed with either *Cd79a-cre* (Cd79a[tm1(cre)Reth]) (33) or *Cd23-cre* (Tg[Fcer2a-cre]5Mbu) (34).

Mice were maintained in the Babraham Institute Biological Support Unit. No primary pathogens or additional agents listed in the FELASA recommendations have been confirmed during health monitoring since 2009. Ambient temperature was ~19–21°C and relative humidity was 52%. Lighting was provided on a 12-h light: 12-h dark cycle including 15 min "dawn" and "dusk" periods of subdued lighting. After weaning, the mice were transferred to individually ventilated cages with one to five mice per cage. Mice were fed CRM (P) VP diet (Special Diet Services) ad libitum and received seeds (e.g., sunflower, millet) at the time of cage-cleaning as part of their environmental enrichment. All mouse experimentation was approved by the Babraham Institute Animal Welfare and Ethical Review Body. Animal husbandry and experimentation complied with existing European Union and United Kingdom Home Office legislation.

## Animal procedures

8–14-wk-old male and female mice were used in this study and experimental cohorts were age- and sex-matched. For generation of bone marrow chimeras, B6.SJL-Ptprc[a]Pepc[b]/BoyJ (B6.SJL-CD45.1) mice were lethally irradiated (2 × 5.0 Gy) and reconstituted with 3 × 10^6 BM cells. Donor BM were composed of the following: 50% B6.SJL *Cd23-cre* and 50% of either *Eif4a1*[+/+] *Cd23-cre* or *Eif4a1*[fl/fl] *Cd23-cre*; or 50% heterozygous (CD45.1+ CD45.2+) and 50% either *Eif4b*[fl/fl] *Eif4h*[fl/fl] *Cd79a*[+/+] or *Eif4b*[fl/fl] *Eif4h*[fl/fl] *Cd79a*[cre/+]. Mice were analysed 10 wk after reconstitution. BrdU (Sigma-Aldrich) was administered at 0.8 mg/ml with 1% sucrose (Thermo Fisher Scientific) in drinking water. For immunisation, mice were injected intraperitoneally with 100 µg NP-KLH (Biosearch Technologies) and precipitated in alum (Universal Biologicals). Mice that had no evidence of a NP-specific response when analysed by flow cytometry or ELISA were excluded from analysis. Administration of substances was performed by a technician blind to the genotype and, where possible, mice with different genotypes were randomised between cages.

## ELISA and ELISPOT

ELISA and ELISPOT were performed as previously published (35, 36) using 20 NP conjugated to BSA for total affinity and 2 NP conjugated to BSA for high affinity. End-point titres were calculated using serial dilution of serum samples.

## In vitro culture and activation of B cells

Irradiated (120 Gy) CD40LB (19) cells were plated at a density of 5 × 10^4 per ml overnight. Splenic B cells were isolated from single cell

suspension using a B cell isolation kit (Cat. # 130-090-862 from Miltenyi). For 24-h analysis, B cells were plated $5 \times 10^5$ per ml in RPMI 1640 Dutch-modified medium (Thermo Fisher Scientific) supplemented with 10% FBS (Gibco), 1 x GlutaMAX (Gibco), 50 $\mu$M 2-mercaptoethanol (Thermo Fisher Scientific), 100 U/ml penicillin and 100 $\mu$g/ml streptomycin (Thermo Fisher Scientific), and 40 ng/ml mIL-4 (214–14; PeproTech). B cells were treated with hippuristanol (gift from Cancer Research Horizons) at the time of plating on CD40LB feeder cells. For puromycin incorporation, cells were treated with 2 $\mu$g/ml puromycin (P8833; Sigma-Aldrich) for final 10 min of culture. For EdU incorporation, cells were treated with 10 $\mu$M ethynyl-2′-deoxyuridine (#E10415; Thermo Fisher Scientific) for final hour of culture. For 72 h CTV analysis isolated B cells were treated for 10 min with 5 $\mu$M CellTrace violet (C34557; Thermo Fisher Scientific) and plated at $7.5 \times 10^4$ cells per ml. Replicates within an experiment are from individual mice, with the number of independent samples indicated.

### Flow cytometry and antibodies

A list of antibodies used is provided in Table S5. Preparation of bone marrow and splenic cells was performed as previously published (35, 37). For intracellular stains, after staining of dead cells and surface markers, cells were fixed with BD Cytofix/Cytoperm (Becton Dickinson) on ice for 30 min, washed in BD perm/wash, and either frozen at –80°C in FCS + 10% DMSO or stained overnight at 4°C in BD perm/wash containing monoclonal rat antibody 2.4G2. EdU was detected with Click-iT Plus EdU AF488 kit (C10632; Thermo Fisher Scientific) as per the manufacturer's instructions. DNA was stained using 20 $\mu$l 7-AAD for 20 min immediately before analysis. Flow cytometry data were acquired on a BD LSRFortessa equipped with five lasers and was analysed using FlowJo software (version 10.7.1).

### Statistical analysis

Statistical analysis was performed using GraphPad. Unless indicated otherwise, $t$ test was used to compare means of two groups with a Holm-Šídák correction test for multiple comparisons. Two-way ANOVA with Tukey's multiple comparison test was used to compare means of more than two groups and two independent variables.

### Sample preparation for mass spectrometry

Splenic B cell isolated from four $Eif4a1^{+/+}$ Cd23-cre or $Eif4a1^{fl/fl}$ Cd23-cre mice were cultured on CD40LB cell for 24 h. Single-cell suspensions were first incubated with 2.4G2 followed by a biotinylated H-2K$^d$ (SF1-1.1) antibody at room temperature in MACS buffer. The biotinylated CD40LB cells were then removed by incubation with anti-biotin microbeads (Militenyi) and passed through an LS column. The purified B cells were washed twice with ice-cold PBS and pellets were frozen on dry ice. A paired $Eif4a1^{+/+}$ Cd23-cre and $Eif4a1^{fl/fl}$ Cd23-cre sample (batch A males) were analysed by mass spectrometry and RNA-seq but removed for differential expression analysis because of reduced cell viability after isolation.

Cell pellets were lysed in 200 $\mu$l of SDS buffer comprised of 5% SDS, 10 mM TCEP, and 50 mM TEAB. Samples were incubated at 95°C for 5 min before sonicating for 15 cycles of 30 s each using a BioRuptor (Diagenode). Proteins were alkylated in the dark for 1 h by the addition of IAA at a final concentration of 20 mM. Protein lysates were prepared for mass spectrometry using s-trap mini columns following the manufacturer's instructions (Protifi). In brief, for each sample 100 $\mu$g of protein was loaded onto an s-trap mini column. Captured protein was washed five times with 400 $\mu$l of wash buffer (90% methanol with 100 mM TEAB, pH 7.1). Proteins were digested by the addition of 5 $\mu$g of trypsin to each sample in 50 mM ammonium bicarbonate. Samples were digested for 3 h at 47°C. Once digestion was complete, peptides were eluted with 80 $\mu$l of 50 mM ammonium bicarbonate followed by 80 $\mu$l of 0.2% formic acid and lastly with the addition of 80 $\mu$l 50% acetonitrile with 0.2% formic acid. After the addition of each elution buffer, columns were centrifuged at 4,000$g$ for 1 min and the flow through collected. Eluted peptides were dried by SpeedVac and suspended in 1% formic acid before quantification using the CBQCA assay (Invitrogen).

### Analysis of peptides by mass spectrometry

Peptides were analysed by data-independent acquisition (DIA) mass spectrometry as described previously (38) For each sample, 1.5 $\mu$g of peptide was injected onto a Q Exactive plus mass spectrometer (Thermo Fisher Scientific) coupled to a Dionex Ultimate 3000 RS (Thermo Fisher Scientific). The following LC buffers were used: buffer A (0.1% formic acid in Milli-Q water [vol/vol]) and buffer B (80% acetonitrile and 0.1% formic acid in Milli-Q water [vol/vol]). An equivalent of 1.5 $\mu$g of each sample was loaded at 10 $\mu$l/min onto a $\mu$PAC trapping C18 column (Pharmafluidics). The trapping column was washed for 6 min at the same flow rate with 0.1% TFA and then switched in-line with a Pharma Fluidics, 200 cm, $\mu$PAC nanoLC C18 column. The column was equilibrated at a flow rate of 300 nl/min for 30 min. The peptides were eluted from the column at a constant flow rate of 300 nl/min with a linear gradient from 1% buffer B to 3.8% buffer B in 6 min, from 3.8% B to 12.5% buffer B in 40 min, from 12.5% buffer B to 41.3% buffer B within 176 min, and then from 41.3% buffer B to 61.3% buffer B in 14 min. The gradient was finally increased from 61.3% buffer B to 100% buffer B in 1 min, and the column was then washed at 100% buffer B for 10 min. Two blanks were run between each sample to reduce carry-over. The column was kept at a constant temperature of 50°C.

Q Exactive Plus was operated in positive ionization mode using an easy spray source. The source voltage was set to 2.2 Kv and the capillary temperature was 275°C. Data were acquired in data independent acquisition mode as previously described (39), with some modifications. A scan cycle comprised of a full MS scan (m/z range from 345-1155), resolution was set to 70,000, AGC target $3 \times 10^6$, maximum injection time was 200 ms. MS survey scans were followed by DIA scans of dynamic window widths with an overlap of 0.5 Th. DIA spectra were recorded at a resolution of 17,500 at 200 m/z using an automatic gain control target of $3 \times 10^6$, a maximum injection time of 55 ms and a first fixed mass of 200 m/z. Normalised collision energy was set to 25% with a default charge state set at 3. Data for both MS scan and MS/MS DIA scan events were acquired in profile mode.

## Proteomics data analysis

Raw mass spec data files were searched as described by reference 40 with some modifications. Raw files were searched using Spectronaut (Biognosys) version 16.0.220606.53000 using the directDIA function. The following search settings were used: minimum peptide length 7, maximum peptide length 52, cleavage enzyme trypsin, maximum missed cleavages 2, protein and peptide FDR was set at 0.01, profiling and cross run normalisation were disabled. Carbamidomethyl (C) was selected as a fixed modification, whereas acetyl (N-term), deamidation (NQ), and oxidation (M) were selected as variable modifications. Data were searched against a hybrid database generated from the UniProt mouse database (June 2020). This hybrid protein database consisted of manually annotated mouse Swiss-Prot entries, along with mouse TrEMBL entries with a manually annotated homologue within the human Swiss-Prot database. Estimated protein copy numbers and concentration were calculated using the proteomic ruler (41) and Perseus (42). Differential expression analysis was performed using the $t$ test function in R after filtering for proteins with a mean > 2 peptides used for quantification across all samples.

## RNA sequencing

RNA was isolated from the purified B cells isolated for mass spectrometry, using an RNeasy Mini Kit (QIAGEN), with on-column DNase treatment, and its quality was assessed on a 2100 Bioanalyser (Agilent). For Illumina sequencing, 100 ng of total RNA was reverse transcribed using NEBNext Ultra II Directional RNA Library Prep Kit for Illumina (#E7760) and NEBNext Multiplex Oligos for Illumina (96 Unique Dual Index Primer Pairs) (#E6440) following the manufacturer's instructions. cDNA quality was checked using Qubit dsDNA HS Assay Kit (Q32854) and Agilent DNA 5000 tapestation reagents (5067-5588; 5067-5589). Samples were pooled at equimolar quantities, and sequenced on a Novaseq 6000 using one lane of an S1 flowcell, as paired end 50-bp (PE50) reads.

## RNA-sequencing data analysis

RNA-sequencing data were trimmed using TrimGalore (v0.6.6; https://www.bioinformatics.babraham.ac.uk/projects/trim_galore/), and quality checked using FastQC (https://www.bioinformatics.babraham.ac.uk/projects/fastqc/) and FastQ Screen (https://www.bioinformatics.babraham.ac.uk/projects/fastq_screen/). Reads were mapped to the GRCm39 mouse genome build using HISAT2 (v2.1.0) (43), suppressing soft-clipping, unpaired or discordant alignments, and taking into account known splice sites from the Ensembl GRCm39 v103 annotation release. TPM were calculated using Stringtie (v2.1.1) (44), using the Ensembl GRCm39 v105 annotation.

For differential expression analysis, raw read counts were generated using Seqmonk (v1.48.2.devel; https://www.bioinformatics.babraham.ac.uk/projects/seqmonk/) over mRNA features with merged isoforms, using the Ensembl GRCm39 v105 annotation, assuming opposing strand-specific, paired reads. DESeq2 (v1.36.0) (45) analysis was performed including experimental batch and sex as batch effects, with "normal" $\log_2$ fold change shrinkage. Significantly, differentially expressed genes were defined as those with FDR-adjusted $P$-value < 0.05 and absolute $\log_2$ fold change > 0.5.

## GSEA

GSEA for gene and protein expression was performed using the GSEA Preranked tool on the GenePattern server, with genes/proteins ranked by $\log_2$-fold change. Gene set used is as follows: early responders, genes with increased expression 2 h after LPS or IgM stimulation (46); late response, genes with increased expression 24 h after IgM stimulation (47); and Mouse MSigDB hallmark gene set (2022).

# Data Availability Statement

The data that support the findings of this study are openly available under the GEO accession GSE237426. The mass spectrometry proteomics data have been deposited to the ProteomeXchange Consortium via the PRIDE (48) partner repository with the dataset identifier PXD046710.

# Supplementary Information

# Acknowledgements

We thank the UKRI-BBSRC Core Capability Grant funded by Babraham Institute Biological Support Unit, Sequencing, Flow Cytometry and Bioinformatics Facilities for support. This study was additionally funded by Blood Cancer UK Grant 14022; Cancer Research Horizons; the BBSRC (BBS/E/B/000C0407; BBS/E/B/000C0428; and a Wellcome Investigator award (200823/Z/16/Z) to M Turner; core funding from Cancer Research UK to the CRUK Beatson Institute (A31287) to OJS, M Bushell, and D Strathdee; OJS laboratory is funded by CRUK (DRCQQR-May21\100002); and work in M Bushell's laboratory is funded by CRUK (A29252).

## Author Contributions

M Screen: conceptualization, data curation, formal analysis, funding acquisition, investigation, methodology, and writing—original draft, review, and editing.
LS Matheson: data curation, formal analysis, and writing—review and editing.
AJM Howden: data curation, formal analysis, methodology, and writing—review and editing.
D Strathdee: resources.
AE Willis: resources.
M Bushell: resources and writing—review and editing.
O Sansom: resources.
M Turner: conceptualization, supervision, funding acquisition, and writing—original draft, review, and editing.

## Conflict of Interest Statement

The authors declare that they have no conflict of interest.

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
