## [Reviewer comments · Life Science Alliance]

Life Science Alliance

RNA helicase EIF4A1-mediated translation is essential for the GC response

Michael Screen, Louise Matheson, Andrew Howden, Douglas Strathdee, Anne Willis, Martin Bushell, Owen Sansom, and Martin Turner

DOI: <https://doi.org/10.26508/lsa.202302301>

Corresponding author(s): *Martin Turner, The Babraham Institute and Michael Screen, Babraham Institute*

Review Timeline:

Submission Date:	2023-07-31
Editorial Decision:	2023-08-21
Revision Received:	2023-10-16
Editorial Decision:	2023-11-03
Revision Received:	2023-11-09
Accepted:	2023-11-09

Transaction Report:

August 21, 2023

Re: Life Science Alliance manuscript #LSA-2023-02301-T

Dr. Martin Turner
The Babraham Institute
Lab. of Lymphocyte Signalling & Dev.
The Babraham Institute
Babraham Hall
Cambridge, UK-Cambridge CB2 4AT CB22 3AT
United Kingdom

Dear Dr. Turner,

Thank you for submitting your manuscript entitled "RNA helicase EIF4A1-mediated translation is essential for the GC response" to Life Science Alliance. The manuscript was assessed by expert reviewers, whose comments are appended to this letter. We invite you to submit a revised manuscript addressing the Reviewer comments.

Thank you for this interesting contribution to Life Science Alliance. We are looking forward to receiving your revised manuscript.

Sincerely,

B. MANUSCRIPT ORGANIZATION AND FORMATTING:

Reviewer #1 (Comments to the Authors (Required)):

In this manuscript, Michael Screen and colleagues investigate the role of RNA helicase EIF4A1 and cofactors EIF4B and EIF4H in B cell development and activation. By using Mb1-Cre- and CD23-Cre-mediated conditional knockout mouse models, the authors show that EIF4A1, but not EIF4B or EIF4H, is essential for early B cell development and germinal center B cell response. Mechanistically, the authors show that EIF4A1 deletion inhibits protein synthesis, MYC expression, and cell cycle regulator expression in activated B cells in vitro. This is an interesting study and the messages are clear.

Major points :

- 1) Eif4a1fl/fl Cd79acre/+ mice have normal numbers of B220+ CD19- (pre-pro) B cells but a drastic decrease in pro-B cells (CD19+ Igμ-) (Figure 1B). What is the specific developmental stage (fraction B, C, C' or D?) of Eif4a1-deficient B cells blocked in Eif4a1fl/fl Cd79acre/+ mice? This information would help to understand the mechanism of EIF4A1 in early B lymphopoiesis.
- 2) GC B cells were almost completely absent in Eif4a1fl/fl Cd23cre/+ mice with a drastic reduction in the number of GC B cells in the spleen compared to EIF4A1-sufficient mice after 7 days of NP-KLH immunization. The authors need to show the DZ/LZ ratio of GC B cells and detect the low/high affinity antibody secreting cells in WT and Eif4a1fl/fl Cd23cre/+ mice, as the data in Fig.3C indicate a significant reduction in high-affinity antibody production in EIF4A1-KO mice.
- 3) The authors have measured MYC expression by intracellular flow cytometry and found a 21% decreased expression in EIF4A1 KO B cells compared to control B cells after activation (IL4+ CD40LB) in vitro. Given that the remaining GC B cells in Eif4a1fl/fl Cd23cre/+ mice had a 40% reduction in EIF4A1 expression compared to GC B cells in Eif4a1fl/fl mice, the authors should determine whether the MYC protein level is downregulated in the same way as in vitro activated B cells.

Additional points :

Fig.1D and Fig.2E, the statistical analysis of the ratios rather than the numbers is much more meaningful.
Lack information on the procedure of puromycin-incorporated protein detection
Line 135, NP-KLH should be NP-KLH

Reviewer #2 (Comments to the Authors (Required)):

The authors present data on the role and biological function of Eukaryotic translation initiation factor 4 A1 (EIF4A) and its cofactors EIF4B and EIF4H in the development of B cells. These factors have been recognized as being critical in several types of cancers, including B cell malignancies, because they promote the translation of mRNAs with structured 5' untranslated regions that occur in several oncogenes. EIF4A and co-factors have ATPase and RNA helicase activity and small molecule inhibitors as potential cancer therapeutics have been developed. The authors summarize in the introduction that in mouse lymphoma models the loss of Eif4a1 or Eif4e leads to improved survival, but that the roles of EIF4A1 and its associated factors EIF4B and EIF4H in B cell development and activation have not been studied. The authors use conditional mouse models, that have previously been established, to inactivate Eif4a1, Eif4b and Eif4h specifically during B cell development and activation. Experimental data are presented that show that EIF4A1 is required for early B cell development and the germinal center (GC) response but not for the maintenance of mature B cells. mRNA translation is normal in EIF4A1-deficient B cells, but a treatment with the EIF4A inhibitor Hippuristanol led to reduced protein synthesis and increased cell death. The authors' final conclusions are that EIF4A1 is required for B cell development and activation.

Major points:

Fig. 1: Eif4a1fl/fl Cd79acre/+ mice are analyzed and Eif4a1fl/fl Cd79acre/+ mice are used as a control. As it is known that Cre alleles can affect the development of normal cells, Eif4a1fl/+ Cd79acre/+ mice would have been a better control. It would be preferable, if the authors showed experiments with this control; or at least provide an example where this control is identical or similar to the Eif4a1fl/fl Cd79a+/+ mice.

Line 86: The authors state that "the remaining pro B cells had only a minor reduction in EIF4A1 expression": are these cells that

escaped the Cre deletion or is the upregulation of EIF4A2 the reason for this? As it stands this conclusion is unclear.

Fig. 2: To appreciate the reduction in expression in the FACS plots in b) an isotype control would be needed; in particular since the authors quantify that: "Fo and MZ B cells had completely lost EIF4A1 expression (Figure 2b) and had a two-fold increase in EIF4A2 expression (Figure 2c)".

Fig. 4: The section between line 165 to 175 is very difficult to follow and needs clarification.

First, the FACS plots in a), d) and e) lack an isotype control. Without this control, it is difficult for instance to appreciate whether the differences in c-MYC expression are significant or only a small variation (despite the p value). A western blot may be difficult here but could help to clarify this. One way to add more experimental evidence for this would be to test c-Myc mRNA levels by RT-PCR to see whether the levels are constant or increased, suggesting that the reduced protein levels are due to decreased translation.

Second, global mRNA translation seems to be decreased in 4A1 Ko cells, but the authors state also on line 173: "However, the expression of MYC in activated 4A1 KO cells was increased three-fold compared to B cells cultured in IL4-only". No data are shown to this effect. This should be clarified either by showing the additional experiments or by text. As it stands it is indeed unclear what happens. Given the importance of c-Myc in GC B cells and B cell lymphoma, the manuscript would gain significantly by clarifying this particular aspect.

Third, does the treatment with Hippuristanol affect c-MYC protein levels?

Fig. 5: The proteomic data are impressive and support indeed the notion of a global reduction of mRNA translation in 4A1 KO B cells. At the same time this argues against a regulation of specific mRNA targets such as c-Myc. This should be discussed. The rationale for the RNA-seq data is unclear. It could be argued that mRNA levels are not affected by EIF4A1 deficiency (only mRNA translation). Thus, any changes in mRNA levels are a consequence of an altered global protein expression profile. Since 4A1 Ko B cells are either eliminated by cell death or stop proliferating, it is somewhat expected to find the mRNA levels of the regulators of these processes changed.

General:

The authors state on lines 73-76: "EIF4A1-deficient naïve B cells have a normal rate of translation but, when treated with the EIF4A inhibitor Hippuristanol, show reduced protein synthesis and increased cell death. Furthermore, we demonstrate that EIF4A1 is essential for the increase in translation following B cell activation".

This needs clarification: Fig. 4d shows decreased levels of puromycin incorporation under CD40LB conditions and the proteomic analysis shows reduced protein expression in 4A1 Ko cells. Hence the statement should be refined. The fact that Hippuristanol reduces global translation is most likely due to its effect on both EIF4A1 and -A2. This should be reworded to make this clearer.

Reviewer #3 (Comments to the Authors (Required)):

This manuscript by Screen et al. investigates the role of EIF4A1 in normal B cell development. Key findings are an essential role for 4A1 at the pro-B and GC stages but not for viability of quiescent naïve B cells. The cofactors 4h and 4b had small effects, only apparent in competitive chimeras. 4A1 ko cells failed to upregulate global protein synthesis, had reduced expression of MYC and cell cycle proteins leading to a proliferation defect. The final conclusion is that 4A1 promotes expression of cells associated with cell cycle. The paper well written, interesting and yields important new findings.

However, there is more the authors could do to tease out global versus specific effects of 4A1 loss. The conclusion states that 4A1 is needed for protein expression of cell cycle proteins. Are the authors suggesting this is a specific effect on translation of these mRNAs? Fig4D shows a failure to upregulate protein synthesis on activation and Fig5B suggests this is a global/general effect.

The authors may be able to make better use of the transcriptomic/proteomic datasets - perhaps plotting fcRNA against fcProtein to see if there is some preferential failure to upregulate cell cycle associated proteins or if there is simply a failure to upregulate any new protein. Are there other protein programs that are preferentially affected? If there is a specific effect on cell cycle associated proteins, how is this specificity mediated?

Some more information about the cell cycle block would be helpful - is there failure to pass a specific cell cycle checkpoint in the 4A1 ko? Is this consistent with the failed upregulation of cell cycle proteins? Would the same block be produced using a global inhibitor of protein translation?

4A2 compensates for 4A1 loss in quiescent cells but is unable to compensate in proliferative states. Is this just a total 4A abundance issue or is there evidence that 4A1 and 4A2 mediate different programs or protein expression?

The MYC stains in Fig4E are interesting because the effect on MYC appears to be relatively modest compared to the huge reduction in the OPP assay. Can the authors speculate why is there such a small effect on MYC - is translation of MYC spared in the absence of 4A1?? Is MCY quantifiable in the proteomic assay?

We thank the reviewers for their constructive comments and have made amendments to improve clarity where it has been suggested.

Reviewer #1

In this manuscript, Michael Screen and colleagues investigate the role of RNA helicase EIF4A1 and cofactors EIF4B and EIF4H in B cell development and activation. By using Mb1-Cre- and CD23-Cre-mediated conditional knockout mouse models, the authors show that EIF4A1, but not EIF4B or EIF4H, is essential for early B cell development and germinal center B cell response. Mechanistically, the authors show that EIF4A1 deletion inhibits protein synthesis, MYC expression, and cell cycle regulator expression in activated B cells *in vitro*. This is an interesting study and the messages are clear.

Major points :

- 1) Eif4a1fl/fl Cd79acre/+ mice have normal numbers of B220+ CD19- (pre-pro) B cells but a drastic decrease in pro-B cells (CD19+ I μ -) (Figure 1B). What is the specific developmental stage (fraction B, C, C' or D?) of Eif4a1-deficient B cells blocked in Eif4a1fl/fl Cd79acre/+ mice? This information would help to understand the mechanism of EIF4A1 in early B lymphopoiesis.

In revised Figure S1 we show the Hardy fractions demonstrating a significant reduction as early as fraction B. Thus, the loss of cells occurred at the earliest stage of B cell development following expression of Cd79-cre.

2) GC B cells were almost completely absent in Eif4a1fl/fl Cd23cre/+ mice with a drastic reduction in the number of GC B cells in the spleen compared to EIF4A1-sufficient mice after 7 days of NP-KLH immunization. The authors need to show the DZ/LZ ratio of GC B cells and detect the low/high affinity antibody secreting cells in WT and Eif4a1fl/fl Cd23cre/+ mice, as the data in Fig.3C indicate a significant reduction in high-affinity antibody production in EIF4A1-KO mice.

In the revised Figure S2 we have included ELISPOT data for IgG1 both at 7 days in the spleen and 28 days in the BM and the LZ/DZ ratio in day 7 splenic GCs. This shows that there are very few IgG1 ASCs detected by a high valency NP capture reagent, which detects both high and low affinity ASC. Even though there is a small shift in ratio to favour LZ GC B cells it is clear the main phenotype is an overall reduction in all GC B cells.

- 2) The authors have measured MYC expression by intracellular flow cytometry and found a 21% decreased expression in EIF4A1 KO B cells compared to control B cells after activation (IL4+ CD40LB) *in vitro*. Given that the remaining GC B cells in Eif4a1fl/fl Cd23cre/+ mice had a 40% reduction in EIF4A1 expression compared to GC B cells in Eif4a1fl/fl mice, the authors should determine whether the MYC protein level is downregulated in the same way as *in vitro* activated B cells.

Only a small fraction (8%) of GC-B cells express MYC *in vivo* and the resolution between MYC negative and positive cells using intracellular flow from *ex vivo* samples is low. Together with the greater than 10-fold reduction in the number of GC-B cells in Eif4a1^{fl/fl} Cd23cre+ mice, with so few cells it would not be possible to get a robust comparison of MYC expression *ex vivo* by intracellular flow cytometry.

Additional points :

Fig.1D and Fig.2E, the statistical analysis of the ratios rather than the numbers is much more meaningful.

Figure 1d and Figure 2e now have statistical analysis of ratios included.

Lack information on the procedure of puromycin-incorporated protein detection.

This information as now been included in the revised methods section on line 350-351

Line 135, NPKLH should be NP-KLH

Reviewer #2

The authors present data on the role and biological function of Eukaryotic translation initiation factor 4 A1 (EIF4A) and its cofactors EIF4B and EIF4H in the development of B cells. These factors have been recognized as being critical in several types of cancers, including B cell malignancies, because they promote the translation of mRNAs with structured 5' untranslated regions that occur in several oncogenes. EIF4A and co-factors have ATPase and RNA helicase activity and small molecule inhibitors as potential cancer therapeutics have been developed. The authors summarize in the introduction that in mouse lymphoma models the loss of *Eif4a1* or *Eif4e* leads to improved survival, but that the roles of EIF4A1 and its associated factors EIF4B and EIF4H in B cell development and activation have not been studied. The authors use conditional mouse models, that have previously been established, to inactivate *Eif4a1*, *Eif4b* and *Eif4h* specifically during B cell development and activation. Experimental data are presented that show that EIF4A1 is required for early B cell development and the germinal center (GC) response but not for the maintenance of mature B cells. mRNA translation is normal in EIF4A1-deficient B cells, but a treatment with the EIF4A inhibitor Hippuristanol led to reduced protein synthesis and increased cell death. The authors' final conclusions are that EIF4A1 is required for B cell development and activation.

Major points:

Fig. 1: *Eif4a1^{fl/fl} Cd79a^{cre/+}* mice are analyzed and *Eif4a1^{fl/fl} Cd79a^{+/+}* mice are used as a control. As it is known that Cre alleles can affect the development of normal cells, *Eif4a1^{fl/+} Cd79a^{cre/+}* mice would have been a better control. It would be preferable, if the authors showed experiments with this control; or at least provide an example where this control is identical or similar to the *Eif4a1^{fl/fl} Cd79a^{+/+}* mice.

Figure R1: Cell number for indicated developing B cell populations using gating strategy from Figure S1a.

We find that *Eif4a1^{fl/+} CD79a^{cre/+}* mice have normal numbers of developing B cells compared to floxed-only mice (**Figure R1**), thus we believe floxed-only mice are a valid control in these experiments.

Line 86: The authors state that "the remaining pro B cells had only a minor reduction in EIF4A1 expression": are these cells that escaped the Cre deletion or is the upregulation of EIF4A2 the reason for this? As it stands this conclusion is unclear.

We expect that these pro B cells have undergone deletion of the genomic region but have not yet degraded all of the EIF4A1 protein. This interpretation has been included on lines 90-92.

Fig. 2: To appreciate the reduction in expression in the FACS plots in b) an isotype control would be needed; in particular since the authors quantify that: "Fo and MZ B cells had completely lost EIF4A1 expression (Figure 2b) and had a two-fold increase in EIF4A2 expression (Figure 2c)".

The use of an isotype control requires the assumption that background staining of the isotype and antibody of interest is the same, but this assumption is frequently incorrect. For intracellular staining, even when the protein of interest is absent in the knockout, the binding of the antibody of interest could be greater than that of the isotype. We have therefore re-worded "Fo and MZ B cells had completely lost EIF4A1 expression" to substantial loss of EIF4A1 expression (line 115) to incorporate the possibility that a low level of protein may remain. EIF4A2 staining was initially validated using *Eif4a2^{fl/fl} CD23cre⁺* mice; this is shown in **Figure R2** as a quality control.

Figure R2: Flow cytometry analysis of EIF4A2 expression in follicular B cells (FoB, CD19+ CD93- CD23+ CD21+). Left: representative flow cytometry plots. Right: Graphs show median fluorescence intensity normalised to T cells (TCRB+).

Fig. 4: The section between line 165 to 175 is very difficult to follow and needs clarification.

First, the FACS plots in a), d) and e) lack an isotype control. Without this control, it is difficult for instance to appreciate whether the differences in c-MYC expression are significant or only a small variation (despite the p value). A western blot may be difficult here but could help to clarify this. One way to add more experimental evidence for this would be to test c-Myc mRNA levels by RT-PCR to see whether the levels are constant or increased, suggesting that the reduced protein levels are due to decreased translation.

For both a) and e) we strongly suggest that the IL4 control, which is a non-proliferating population, identified the baseline MFI of a CD69-negative or MYC-negative population with which difference in expression can be appreciated. We agree a baseline for puromycin expression is important. In our experiment, IL4 control cells are still synthesising protein. However, inhibiting translation is likely to be a better control for setting a baseline; figure 4H shows this reduction in translation in both activated and IL4 only culture conditions. We have also included an example where 24-hour IgM-stimulated B cells were treated with Hippuristanol 10 minutes prior to the addition of puromycin (**Figure R3**), which highlights that the baseline for these experiments is very low.

We do accept that the difference in MYC expression that we show is small, but we do not believe Western blots, which are difficult to quantitate accurately, would quantitate this difference any more accurately than flow cytometry. We have however modified the text in this section (as suggested in the second point, below) to afford a greater clarity.

Figure R3: B cells from B6.SJL mice cultured in IL4, with CD40LB-expressing feeder cells or IgM for 24 hours. Cells treated with puromycin in final 10 minutes of culture. 2µM Hippuristanol (Hipp) add 10 minutes before the addition of puromycin.

Second, global mRNA translation seems to be decreased in 4A1 Ko cells, but the authors state also on line 173: "However, the expression of MYC in activated 4A1 KO cells was increased three-fold compared to B cells cultured in IL4-only". No data are shown to this effect. This should be clarified either by showing the additional experiments or by text. As it stands it is indeed unclear what happens. Given the importance of c-Myc in GC B cells and B cell lymphoma, the manuscript would gain significantly by clarifying this particular aspect.

As mentioned above we have reordered and edited this section to make it clearer. In figure 4E we show that MYC MFI in 4A1 KO B cells increases from 342 in IL4 only conditions to 965 when activated with 40LB cells; thus, we took that as a three-fold increase in MYC expression. We have now reworded this as a three-fold increase in MFI following stain with a MYC antibody (line 185-186).

Third, does the treatment with Hippuristanol affect c-MYC protein levels?

Figure R4: MYC expression in B cells from B6.SJL mice treated with different doses of Hippuristanol

400nM Hippuristanol reduced MYC expression in CD40LB activated B cells to the level of IL4 only conditions (Figure R4). However, when cells are cultured with 80nM Hippuristanol MYC expression is not significantly reduced, but puromycin incorporation was reduced by more than 3-fold (Figure 4h). This suggests that, in activated B cells, MYC can be translated even when EIF4A activity is diminished.

Fig. 5: The proteomic data are impressive and support indeed the notion of a global reduction of mRNA translation in 4A1 KO B cells. At the same time this argues against a regulation of specific mRNA targets such as c-Myc. This should be discussed.

We agree that the data supports a more global effect on translation rather than a mRNA specific effect and have therefore included more discussion on this. We believe the discussion in relation to MYC is still required due to its importance to selection in GCs and previous reports linking EIF4A1 and MYC expression.

The rationale for the RNA-seq data is unclear. It could be argued that mRNA levels are not affected by EIF4A1 deficiency (only mRNA translation). Thus, any changes in mRNA levels are a consequence of an altered global protein expression profile. Since 4A1 Ko B cells are either eliminated by cell death or stop proliferating, it is somewhat expected to find the mRNA levels of the regulators of these processes changed.

It is important to highlight that in our experiments the point of analysis is prior to the first cell division and that the cells are not dying (Figure 4c). RNA abundance, which we assume shouldn't be directly altered by changes in EIF4A1 expression, can be used to infer if and how the activation of 4A1 KO cells differs from the WT. Differences are evident for transcripts that encode cell cycle proteins such as cyclins and cyclin-dependent kinases, which has now been included in the revised Figure 5 in response to reviewer 3.

General:

The authors state on lines 73-76: "EIF4A1-deficient naïve B cells have a normal rate of translation but, when treated with the EIF4A inhibitor Hippuristanol, show reduced protein synthesis and increased cell death. Furthermore, we demonstrate that EIF4A1 is essential for the increase in translation following B cell activation".

This needs clarification: Fig. 4d shows decreased levels of puromycin incorporation under CD40LB conditions and the proteomic analysis shows reduced protein expression in 4A1 Ko cells. Hence the statement should be refined. The fact that Hippuristanol reduces global translation is most likely due to its effect on both EIF4A1 and -A2. This should be reworded to make this clearer.

This section has been reworded for greater clarity.

Reviewer #3

This manuscript by Screen et al. investigates the role of EIF4A1 in normal B cell development. Key findings are an essential role for 4A1 at the pro-B and GC stages but not for viability of quiescent naïve B cells. The cofactors 4h and 4b had small effects, only apparent in competitive chimeras. 4A1 ko cells failed to upregulate global protein synthesis, had reduced expression of MYC and cell cycle proteins leading to a proliferation defect. The final conclusion is that 4A1 promotes expression of cells associated with cell cycle. The paper well written, interesting and yields important new findings.

However, there is more the authors could do to tease out global versus specific effects of 4A1 loss. The conclusion states that 4A1 is needed for protein expression of cell cycle proteins. Are the authors suggesting this is a specific effect on translation of these mRNAs? Fig4D shows a failure to upregulate protein synthesis on activation and Fig5B suggests this is a global/general effect.

We agree the puromycin incorporation experiments and proteomics would be in keeping with a global reduction in translation in 4A1 KO cells. However, it is clearly not a failure to increase all protein production. This is highlighted by the increased expression of CD69 (Figure 4a) and MYC (Figure 4e) after B cell activation, as shown by flow cytometry. We have further included BCLXL (Figure 4d) as a protein that is increased after activation in both control and 4A1 KO B cells. We have also included more discussion on the potential of global versus specific effects as a response to reviewer 2.

The authors may be able to make better use of the transcriptomic/proteomic datasets - perhaps plotting fcRNA against fcProtein to see if there is some preferential failure to upregulate cell cycle associated proteins or if there is simply a failure to upregulate any new protein. Are there other protein programs that are preferentially affected? If there is a specific effect on cell cycle associated proteins, how is this specificity mediated?

Given that the B cells at 24 hours after activation are not at steady state (their transcriptomes and proteomes are highly dynamic) simply plotting fcRNA against fcProtein at a global level could be misleading and unlikely to provide sufficient evidence for EIF4A1-specific effects. To investigate whether specific protein programs are affected and to understand how this specificity is mediated would require techniques that analyse mRNA translation integrated with the identification of

transcripts bound by EIF4A1, potentially at multiple timepoints. We believe this significant body of data is beyond the scope of this manuscript.

Some more information about the cell cycle block would be helpful - is there failure to pass a specific cell cycle checkpoint in the 4A1 ko? Is this consistent with the failed upregulation of cell cycle proteins?

We have further characterised the cell cycle block and included the data in (Figure 5f) This shows that 4A1 KO B cells have a reduced proportion of S-phase cells 24 hours after activation. This is consistent with the analysis of the transcriptomics and proteomics.

Would the same block be produced using a global inhibitor of protein translation?
Treatment of CD40LB activated B cells with 400nM Hippuristanol shows that a global inhibitor of protein translation kills the cells prior to the first cell division (Figure 4g).

4A2 compensates for 4A1 loss in quiescent cells but is unable to compensate in proliferative states. Is this just a total 4A abundance issue or is there evidence that 4A1 and 4A2 mediate different programs or protein expression?

In naïve cells that are quiescent EIF4A1 and EIF4A2 have 1.3×10^5 and 6.5×10^4 copies respectively and, from the same dataset, BCR-activated B cells have 4.4×10^6 and 2.9×10^5 copies respectively (<http://immpres.co.uk/>). Thus, EIF4A1 makes up the vast amount of total EIF4A in activated B cells. It is therefore possible the phenotype in activated 4A1 KO B cells is due to reduced total EIF4A abundance because, unlike in naïve B cells, the three-fold increased EIF4A2 expression would be insufficient to restore EIF4A function. This is contrary to a previous report, which showed that EIF4A2 was unable to compensate for the loss of EIF4A1, even when total EIF4A levels remained the same (ref 12, PMID: 22589333).

The MYC stains in Fig4E are interesting because the effect on MYC appears to be relatively modest compared to the huge reduction in the OPP assay. Can the authors speculate why is there such a small effect on MYC - is translation of MYC spared in the absence of 4A1?? Is MYC quantifiable in the proteomic assay?

MYC was not detected in our proteomics dataset, but its transcript levels remain unchanged between 24-hour activated B cells from *Eif4a1^{fl/fl}* and *Eif4a1^{fl/fl} Cd23cre+* mice. Given the small change in MYC expression between control and 4A1 KO activated B cells, as measured by flow cytometry, we suggest that significant amounts of MYC can be translated and accumulate independently of EIF4A1. Furthermore, as 400nM Hippuristanol treatment reduces the MYC MFI in activated B cells down to that present in IL4 only cells (Figure R4) we can speculate that translation of MYC in the EIF4A1 KO B cell is mediated by EIF4A2. However, we cannot rule out the possibility that MYC may be translated by EIF4A-independent mechanisms.

November 3, 2023

RE: Life Science Alliance Manuscript #LSA-2023-02301-TR

Dr. Martin Turner
The Babraham Institute
Lab. of Lymphocyte Signalling & Dev.
Cambridge CB22 3AT
United Kingdom

Dear Dr. Turner,

Thank you for submitting your revised manuscript entitled "RNA helicase EIF4A1-mediated translation is essential for the GC response". We would be happy to publish your paper in Life Science Alliance pending final revisions necessary to meet our formatting guidelines.

- please add the Twitter handle of your host institute/organization as well as your own or/and one of the authors in our system
- please move your figure legends after the References section
- please add callouts for Figures 4H and 5E, H to your main manuscript text

A. FINAL FILES:

B. MANUSCRIPT ORGANIZATION AND FORMATTING:

****It is Life Science Alliance policy that if requested, original data images must be made available to the editors. Failure to provide**

original images upon request will result in unavoidable delays in publication. Please ensure that you have access to all original data images prior to final submission.**

The license to publish form must be signed before your manuscript can be sent to production. A link to the electronic license to publish form will be available to the corresponding author only. Please take a moment to check your funder requirements.

Sincerely,

Reviewer #1 (Comments to the Authors (Required)):

The authors have adequately addressed my concerns.

Reviewer #2 (Comments to the Authors (Required)):

The manuscript addresses the role of eukaryotic translation initiation factors 4A1 (EIF4A1) and its associated factors EIF4B and EIF4H in B cell development and activation, for which only limited information exists to date. This is experimentally addressed using mouse models that enable the conditional inactivation of the Eif4a1, Eif4b and Eif4h genes during B cell development and activation. The authors present data to support several new claims: first, that EIF4A1 plays an important role in defined early developmental stages of B cells and later in the germinal center. Second, evidence is presented demonstrating that B cells need EIF4A1 to adapt their rate of mRNA translation upon activation, regulate MYC expression and the expression of cell cycle regulators. In summary, this paper reveals new and useful information how B cell development and activation are controlled at the level of mRNA translation. The first version of the manuscript has been carefully revised by the authors responding and addressing all concerns of the reviewers. No further revisions are required.

Reviewer #3 (Comments to the Authors (Required)):

The authors have now responded to my comments with new data, text and clarifications. I am happy that the manuscript identifies important new understanding and that the conclusions are supported by the data. I support publications of the manuscript as is.

November 9, 2023

RE: Life Science Alliance Manuscript #LSA-2023-02301-TRR

Dr. Martin Turner
The Babraham Institute
Lab. of Lymphocyte Signalling & Dev.
Cambridge CB22 3AT
United Kingdom

Dear Dr. Turner,

Thank you for submitting your Research Article entitled "RNA helicase EIF4A1-mediated translation is essential for the GC response". It is a pleasure to let you know that your manuscript is now accepted for publication in Life Science Alliance. Congratulations on this interesting work.

DISTRIBUTION OF MATERIALS:

Again, congratulations on a very nice paper. I hope you found the review process to be constructive and are pleased with how the manuscript was handled editorially. We look forward to future exciting submissions from your lab.

Sincerely,
